# Molecular Docking of Natural Compounds for Potential Inhibition of AhR

**DOI:** 10.3390/foods12101953

**Published:** 2023-05-11

**Authors:** Deborah Giordano, Angelo Facchiano, Stefania Moccia, Anna Maria Iole Meola, Gian Luigi Russo, Carmela Spagnuolo

**Affiliations:** National Research Council, Institute of Food Sciences, 83100 Avellino, Italy; deborah.giordano@isa.cnr.it (D.G.); angelo.facchiano@isa.cnr.it (A.F.); stefania.moccia@isa.cnr.it (S.M.); iolemeola@hotmail.com (A.M.I.M.); glrusso@isa.cnr.it (G.L.R.)

**Keywords:** aryl hydrocarbon receptor, molecular docking, phytochemicals, benzo[a]pyrene

## Abstract

The aryl hydrocarbon receptor (AhR) is a highly conserved environmental sensor, historically known for mediating the toxicity of xenobiotics. It is involved in numerous cellular processes such as differentiation, proliferation, immunity, inflammation, homeostasis, and metabolism. It exerts a central role in several conditions such as cancer, inflammation, and aging, acting as a transcription factor belonging to the basic helix–loop–helix/Per-ARNT-Sim (bHLH-PAS) protein family. A key step in the canonical AhR activation is AhR-ARNT heterodimerization followed by the binding to the xenobiotic-responsive elements (XREs). The present work aims to investigate the potential AhR inhibitory activity of selected natural compounds. Due to the absence of a complete structure of human AhRs, a model consisting of the bHLH, the PAS A, and the PAS B domains was constructed. Blind and focused docking simulations revealed the presence of further binding pockets, different from the canonical one presented in the PAS B domain, which could be important for AhR inhibition due to the possibility to impede AhR:ARNT heterodimerization, either preventing conformational changes or masking crucial sites necessary for protein–protein interaction. Two of the compounds retrieved from the docking simulations, i.e., β-carotene and ellagic acid, confirmed their capacity of inhibiting benzo[a]pyrene (BaP)-induced AhR activation in in vitro tests on the human hepatoma cell line HepG2, validating the efficacy of the computational approach.

## 1. Introduction

The aryl hydrocarbon receptor (AhR) is a highly conserved environmental sensor, mainly known for its key role in the metabolism of xenobiotic ligands [1]. It is the mediator of the toxic effects of exogenous and endogenous metabolites such as environmentally persistent dioxins [2,3]. In the last 20 years, the growing scientific interest toward AhR transcriptional activity has been correlated to its role in health and diseases [4]. The canonical AhR signaling pathway is triggered in response to polycyclic aromatic hydrocarbons (PAHs), such as benzo[a]pyrene (BaP), and results in the activation of CYP450 family members. The non-canonical AhR pathways involve several crucial signaling mediators such as JNK, Wnt, HIF-1, SRC, and TGF-β and interfere with the modulation of numerous cellular processes such as differentiation, proliferation, immunity, inflammation, homeostasis, and metabolism [5].

The AhR structure is composed of 848 residues assembled in four domains: the N-terminal basic helix–loop–helix (bHLH) domain (amino acids from 27 to 80), two Per-ARNT-Sim (PAS) domains (A, from 111 to 181, and B, from 275 to 342), and the carboxy (C-) terminal transactivation domain (TAD) (from 490 to 805) [6]. A PAC repeat region follows the PAS B domain from the amino acid 348 to 386; this motif is proposed to contribute to the PAS domain fold [7]. The bHLH domain of the AhR mediates the formation of protein homodimeric complexes and heterodimerization with the bHLH-PAS factor ARNT necessary for DNA binding and transcriptional activation of target genes [8]. The PAS A domain is required for both dimerization and heterodimerization with the PAS domain of the AhR and ARNT, respectively; therefore, it controls the transcriptional activity [9]. The PAS B domain represents the ligand binding domain and probably possesses a regulatory function without direct involvement in the dimerization [6]. The TAD is capable of sequestering coactivators/cofactors involved in transcriptional activity with the different ability of inhibiting the AhR depending on each specific subdomain (acidic, Q-rich, and PST-rich subdomains) composing its structure [10].

The AhR is a member of the class I basic helix–loop–helix/PAS family of nuclear transcription factors. In solution and in crystal, the AhR PAS A domain is found in a helix-swapped homodimeric assembly; however, until now, no evidence has been provided on the existence of a physiological role of a full-length AhR as a homodimer [9]. In contrast, its ability to interact with members of the bHLH-PAS class II is well known, in particular, with the aryl hydrocarbon receptor nuclear translocator (ARNT). The AhR is located in the cytoplasm as a heterodimeric complex association with two members of the Hsp90 family, p23 and XAP2. Hsp90 interacts via both the bHLH region and PAS B, and probably masks its nuclear localization sequence (NLS). Ligand binding results in a conformational change, which exposes the NLS, and the entire complex can then translocate into the nucleus where the partner protein ARNT displaces proteins bound to the AhR. The AhR/ARNT heterodimer interacts with the target genes (such as CYP1A1) in the upstream enhancer region of the xenobiotic-responsive element, recruits transcriptional coactivators through transactivation domains (TADs) of both the ARNT and AhR, and activates the transcription [11,12].

Considering the crucial role of the AhR, an extremely interesting field of investigation is the identification of either AhR agonists or antagonists able to modulate their contribution to physiological and pathophysiological processes. Among possible AhR ligands, many pieces of the literature indicate that natural bioactive compounds may represent potential candidates. Polyphenols are among the most studied natural compounds. In particular, some flavonoids belonging to the flavone, flavonol, flavanone, and isoflavone subclasses have been indicated as AhR modulators [13]. These data were derived mainly from in vitro studies where, using a wide range of concentrations (10–100 µM), agonist and/or antagonist activities were described. Similarly, other classes of natural compounds seem to modulate the AhR and include carotenoids, such as lycopene or β-carotene [14,15].

Based on these concepts, the present work aims to define the presence of binding pockets, different from the canonical ones, involved in the AhR:ARNT heterodimerization and, through molecular docking simulations, evaluate the inhibitory potential of selected natural compounds.

## 2. Materials and Methods

### 2.1. Reagents

Dulbecco’s Modified Eagle’s Medium (DMEM) (Euroclone, Milan, Italy), fetal bovine serum, L-glutamine 200 mM, penicillin 5000 IU/mL/streptomycin 5000 g/mL, non-essential amino acids (NEAAs), and PBS (phosphate buffer saline) were purchased from Thermo-Fisher Scientific/Life Technologies (Milan, Italy). The β-carotene, ellagic acid, curcumin, and dimetylsulfoxide (DMSO) were from Merck Life Science (Milan, Italy). Benzo[a]pyrene was kindly donated by Dr. Maria Grazia Volpe.

### 2.2. AhR Model Construction

Due to the absence of a crystallographic structure for the AhR available in the Protein Data Bank (PDB) (accessed on 6 June 2022 https://www.rcsb.org/), a 3D structure of the first half of the sequence was modeled. It was possible to build two models: one for the AhR region from amino acids 32 to 388 including the first three domains (bHLH, PAS A, and PAS B until the PAC motif) and the second for the PAS B domain (from 286 to 394) only.

Templates used for the construction of the multidomain protein were PDB structures 4H6J and 3F1P for the PAS B domain, 4M4X, 5NJ8, and 5Y7Y for the bHLH/PAS A domains, and 4ZPZ as template for all three and their connection. The 4H6J chain A was a human hypoxia inducible factor 1-alpha, and its amino acid sequence had 31% of identity with AhR sequence (UniProt code: P35869). The 3F1P chain A was the human endothelial PAS Domain-containing Protein 1 and shared 27% of identity. The 4M4X chain A and 5NJ8 chain C were the AhR PAS A domains from *Mus musculus* (83% of sequence identity) and *Homo sapiens*. The 5Y7Y chain A was the structure of the human AhR repressor with 55% of sequence identity. The best template in terms of coverage was represented by the endothelial PAS Domain-containing Protein 1 from *Mus musculus* (4ZP4 chain B); however, it showed only 25% of sequence identity. The presence of multiple templates was necessary in order to balance templates with a higher percentage of sequence identity but was penalized by the presence of multiple unresolved portions in the structure.

The model built only for PAS B domain was performed exploiting not only 4H6J and 3F1P, but also 4M4X, whose alignment with the AhR sequence was optimized no longer for the PAS A but for the first part of the PAS B, and 4F3L chain A (circadian locomotor output cycles protein kaput from *Mus musculus*), which showed the best coverage despite the low sequence identity (24%).

All templates downloaded from PDB were validated in terms of resolution, missing atoms/residues, percentage of query coverage, and sequence identity. In the same way, all models built by Modeller 9.19 were validated in order to select the best one considering all the structural features. Particularly, Z-score, Q-mean, and Ramachandran plot were considered, as determined by ProSA-web [16], QMEAN-SWISS-MODEL [17], and PROCHECK [18], respectively.

### 2.3. Docking Analysis

Ligand structures were downloaded in sdf format from PubChem database [19] and converted in pdb format by Chimera (www.rbvi.ucsf.edu/chimera, accessed on 6 June 2022). Ligand and protein structures were prepared for the docking by AutoDock Tool 1.5.6 [20]. Blind and focused docking simulations were performed using, for the first strategy, the model obtained for the first three domains of the AhR and, for the second, the model obtained for the PAS B domain. RMSD among the PAS B portion of the multidomain model and PAS B model was equal to 0.480 Å with a perfect fitting of all the secondary structure elements. However, the choice to use two different models was due to the optimization of the multidomain interactions and the PAS A composition of the first, more indicated for blind docking, and the single domain optimization reached by the second, perfect for the focused approach. Docking simulations were executed using AutoDock 4.2.5.1 [21], and results were analyzed in terms of ligand interactions, pockets detected, and comparison between blind and focused results, using AutoDock Tool 1.5.6.

### 2.4. Cell Culture and Treatments

Hepatoma cell line, HepG2, purchased from ATCC (Manassas, VA, USA), was cultured in DMEM supplemented with 10% fetal bovine serum, 1% L-glutamine, 1% penicillin/streptomycin, and 1% NEAA at 37 °C in a humidified atmosphere containing 5% CO_2_. Before each experiment, cells were counted with the Trypan blue dye using the EVE Automatic cell counter (NanoEntek distributed by VWR, Milan, Italy) to assess their viability (usually >90%). Cells (1 × 10^6^) were pre-treated for 1 h with 1 μM of natural compounds (β-carotene, curcumin, ellagic acid); subsequently, 5 μM BaP was added for 15 h. To perform the experiments, curcumin and ellagic acid were dissolved in DMSO. In order to prepare an oil-in-water (o/w) nanoemulsion, β-carotene dissolved in n-hexane (oil-phase) was dispersed in an aqueous phase [22]. Briefly, β-carotene (0.03% *w*/*v*) was dissolved in n-hexane before adding in a 1:9 ratio to the aqueous phase containing Tween 20 (0.5%) as the emulsifier. The emulsions were homogenized using an Ultra-Turrax homogenizer (T8, Ika-Werke, Stauffen, Germany) performing three cycles at 10,000× *g* for 2 min to form a coarse emulsion, followed by sonication (Sonicator, ultrasonic processor XL, Misonix, NY, USA) to further reduce the particle size. Subsequently, the solvent was removed from nanoemulsions under nitrogen vapors before membrane filtration to sterilize samples throughout a 0.2 μm membrane. Control nanoemulsions (NEs) were prepared without β-carotene in the n-hexane oil phase in the aqueous solution. The particle size of β-carotene nanoemulsions was determined using dynamic light scattering. Briefly, particle size (expressed as “derived diameter”) distributions were measured by LALLS (low-angle laser light scattering technique), confirming the nanometer size of the particle diameter. The concentration of β-carotene incorporated into the nanoemulsion was determined by extracting 0.5 mL of nanoemulsion with a mixture of n-hexane and absolute ethanol (2:1, *v*/*v*) before the measurement of the amount of β-carotene using spectrophotometric and/or HPLC analysis. The stability of the nanoemulsion was evaluated over time by measuring the β-carotene content for up to 3 months using spectrophotometric and/or HPLC analysis. The nanoemulsion proved to be stable without the separation of the two phases and changes in the β-carotene amount. The vehicle, DMSO, was applied at a final concentration of 0.1% (*v*/*v*), which resulted as being not cytotoxic on HepG2 cells. Equal volumes of solutions (natural compounds and vehicles) were added to the cells.

### 2.5. RNA Extraction, Retro Transcription, and Real-Time PCR Quantification

Treated HepG2 cells were lysed and total RNA was extracted using SV Total RNA Isolation System (Promega Corporation, Milan, Italy) according to the manufacturer’s protocol. RNA quantification was assessed with Synergy HT microplate reader (BioTek, Milan, Italy). Reverse transcription (RT) for cDNA synthesis was performed on 1 µg total RNA using the LunaScript^®^ RT SuperMix Kit (New England BioLabs, Euroclone, Pero, Milan, Italy). Real-time PCR analysis was performed using the Power SYBR Green PCR Master Mix (Applied Biosystems, Monza, Italy) according to the manufacturer’s protocol with a 7000 Real-Time PCR System (Applied Biosystems). Quantification was performed in duplicate or triplicate, and expression levels of CYP1A1 (forward-GATTGAGCACTGTCAGGAGAAGC, reverse-ATGAGGCTCCAGGAGATAGCAG) were normalized using internal standards: GAPDH (forward-CCACTAGGCGCTCACTGTTCT, reverse-GCGAACTCACCCGTTGACT). Relative gene expression levels corresponded to fold induction (2^−ΔΔCt^) compared with untreated cells. Significant differences were determined using Student’s *t*-test.

## 3. Results

### 3.1. AhR Model

Due to the complexity of the multidomain structural organization of the AhR, as reported in the Methods section, it was decided to build two models for the AhR structure. The first included the first three domains: bHLH, PAS A, and PAS B until the PAC motif (Figure 1A); the second included only the PAS B domain (Figure 1B) by exploiting different templates.

Models obtained showed acceptable quality parameters (see Appendix A). Despite the availability of a model of the entire protein from AlphaFold [23], we adopted these two simulation models based on the quality of details retrieved, as discussed below (see Discussion section). In Figure 2, we show the model of the whole protein as available in the AlphaFold archive and the comparison with our models. In brief, while the overlap of the structures demonstrates that the secondary structure elements are very similarly assembled in the domain architecture, the relative assembly of the domains is different; therefore, we considered our models more appropriate for the aims of the present work.

### 3.2. Docking Results

To detect additional possible binding pockets that could be present on the AhR beyond the canonical one in the PAS B domain, blind docking was performed using TCDD (2,3,7,8-tetrachlorodibenzo-p-dioxin) and BaP, already known for being high-affinity ligands. Moreover, focused docking on these molecules was performed in order to detect their mean binding energy in the canonical pocket, necessary for a reliable comparison with energy results obtained by the additional docking performed with ligands of interest (Table 1).

From the collection and the analysis of all the docking simulations results, it was possible to detect four pockets on the AhR structure relevant for its inhibition (for simplicity, we will refer to them as A–B–C–D; Figure 3).

Pocket A involves mainly amino acid residues located on the PAS A domain, two residues of the PAS B (Q273-I280) domain, and one involved in the ARNT binding area (N123). Pocket B involves both PAS A and PAS B residues and again N123, located at the ARNT binding area. Both these pockets could be useful for AhR inhibition because a possible binding in these areas could move the PAS A and B domains closer, stabilizing a close conformation of the receptor. This entails a reduction in the surface available for the correct ARNT binding due to the steric hindrance of the PAS B domain. Therefore, a binding in these regions can be translated into an inhibition of the heterodimerization crucial for AhR activation.

Pocket C is the canonical pocket in PAS B known for being the xenobiotic binding site; a binding on this site can induce an activation of the receptor.

Pocket D is the interaction area between AhR and ARNT (E114; F117-V126; Y137; F266-I268); residues involved in this pocket have been experimentally described [6] as being crucial for ARNT:AhR dimerization. A binding in this area is potentially able to inhibit the heterodimerization engaging residues responsible for ARNT binding in the interactions with the ligand.

We selected, among different classes, some natural compounds which, based on the literature data [13,14,15], would have been able to modulate the AhR signaling pathway or to inhibit the unhealthy effects of PAHs. The literature search was conducted on different websites (Pubmed; Google Scholar; PubChem; Drugbank; FoodB; Phenol Explorer) using “compound” + “AhR” + ”PAHs” + “TCDD” + “BaP” as keywords. The detection of the relevant pockets was followed by the analysis of the docking results, related to the 14 hypothetical ligands tested, and their ranking according to the binding energy values, the number of poses in each cluster, and the location of the binding to the AhR multidomain model (Table 2)

The whole description of the results obtained for blind and focused docking analyses, from which the ranking was retrieved, was reported in the Appendix A.

The obtained results reported in Table 2 highlight how β-carotene, lycopene, xanthohumol, hesperetin, and naringenin appear to be the most promising candidates, even with a different degree which justifies their order of ranking from the first to the fifth. In particular, β-carotene is selected as the best one due to its optimal binding energy and its high selectivity for pocket D, responsible for the inhibition of the heterodimerization AhR:ARNT. The β-carotene binding energy to this pocket is detected in the best pose (LBE) as equal to −12.57 kcal/mol. Moreover, the MBE calculated on the 32 poses that compose the cluster is equal to −10.65 kcal/mol, a value which is almost 1 kcal/mol lower than the best binding energy (BaP binding) in the canonical pocket (−9.33 kcal/mol). Figure 4 shows that β-carotene binding involves almost half of the residues responsible for the interaction with ARNT, which results in being hidden by its presence. Moreover, β-carotene also makes direct interaction with four of the fifteen residues that actively bind ARNT: V126-Y137-F266-A267, and it is undeniable that its presence represents a physical hindrance to ARNT binding.

Lycopene seems to be slightly lower selective than β-carotene, as demonstrated by the detection of two different binding sites: one between pockets A and B, and the second in pocket D. For both, the mean binding energy is comparable to BaP and TCDD binding energy, indicating a possible efficient binding to the receptor. Both the binding zones interested by lycopene could be useful for AhR inhibition. Xanthohumol presents a fair amount of clusters, each one composed of few poses, which show its binding position between pockets A and B. Energies are variable but the lowest mean binding energy (−9.05 kcal/mol) is comparable to BaP mean binding energy. A direct inhibition of the heterodimerization seems to be possible also in this case even if the binding in pocket D shows higher energy.

The case of hesperetin is different. Here, the ligand detects three binding sites, one of which is represented by pocket C, suggesting also a possible activation role. Pockets A and D show a higher selectivity than pocket C with a number of poses in clusters four times higher than in pocket C and the lowest binding energy around −8.00 kcal/mol. Similarly, in this case, results suggest a possible inhibitory role of hesperetin.

Naringenin is probably the last compound for which it is possible to hypothesize an inhibitory role. It detects all four pockets on the AhR multidomain model, with a marked selectivity for pocket A. In this site, the lowest binding energy is still near the value of −8.00 kcal/mol; however, for all the other pockets, despite a good selectivity proven by the number of poses in clusters, the energy is drastically increased.

For curcumin and quercetin, borderline results were obtained. In fact, even if they seem to present some values of binding that are quite comparable with the ones described for naringenin, in particular the binding between pockets A and B for curcumin (lowest MBE = −7.87 kcal/mol) and in pocket A for quercetin (lowest MBE = −7.46 kcal/mol), they present low selectivity. Curcumin shows, in the first three docking results, more of an affinity for pockets not involved in the inhibition (Appendix A). Instead, quercetin presents a higher selectivity for one of the pockets not useful for the inhibition, detected also by curcumin, for which the binding energy is quite comparable to those useful (MBE = −6.84 kcal/mol; LBE = −7.38 kcal/mol) (Appendix A).

Results related to resveratrol, genistein, and epigallocatechin-3-gallate underline binding poses that are equally distributed between relevant and irrelevant sites with a major propensity for the binding to pockets that are not relevant (Supplementary Appendix A).

Cyanidin-3-glucoside, hydroxytyrosol, and sulforaphane show high binding energy; therefore, they are considered not compatible with an inhibitory activity (Appendix A).

A special case may regard the ellagic acid, ranked in position 13, not for its low capacity to induce inhibition but for the possibility to activate the receptor. The docking results retrieved for this compound are not easy to interpret. Ellagic acid seems to have a possible inhibitory role in binding the area between pockets A and B with a strong selectivity (30 poses in the cluster) and a discrete binding energy (MBE = −6.77 kcal/mol; LBE = −7.28 kcal/mol). This result is followed by a number of additional clusters whose poses are still compatible with a possible inhibitory role. Pocket C is present as the eleventh cluster in the blind docking rank, with an MBE equal to −6.23 kcal/mol and 12 poses in the cluster, suggesting a possible activation role at high concentrations. Nevertheless, the focused docking on PAS B data (Appendix A) shows strong selectivity for the external portion of PAS B rather than for the canonical C pocket, invalidating the hypothesis of its role as an activator.

The controversial behavior of ellagic acid makes it of high interest for experimental validation together with β-carotene, which resulted as the most credible inhibitor candidate, and curcumin for its intermediate docking data.

### 3.3. Experimental Validation

The HepG2 cell line expresses the AhR and it is commonly employed to study the biological effects mediated by this factor [24,25]. For this reason, we used HepG2 cells to validate in vitro the results of the docking screening. Among the natural compounds previously analyzed, we decided to investigate the capacity of β-carotene, curcumin, and ellagic acid to inhibit the activation of the AhR pathway, as reported below.

The level of the CYP1A1 mRNA target gene, a prototypical biomarker used as an indicator of AhR activation [26], was evaluated after cell stimulation. In particular, we pre-treated HepG2 cells for 1 h with 1 μM concentrations of the selected compounds, followed by treatment with BaP 5 μM. We decided to use concentrations of natural compounds, compatible with their low bioavailability. As reported in Figure 5A, the β-carotene treatment significantly reduced the strong increment of CYP1A1 expression measured after BaP stimulation, as expected. Gene expression was not modified using the control nanoemulsion (NE). Differently, in Figure 5B,C, we observed that curcumin did not exert any inhibitory effect against BaP, while ellagic acid, surprisingly, was able to significantly reduce the CYP1A1 expression with respect to BaP induction.

## 4. Discussion

The AhR is a transcription factor historically known as a mediator of xenobiotics toxicity; however, a central role of the AhR in several pathophysiological conditions, such as cancer, inflammation, and aging, emerged from studies in the last decade suggesting new therapeutic opportunities based on targeting its functions. A key step in the canonical AhR activation is AhR-ARNT heterodimerization followed by binding to xenobiotic-responsive elements (XREs); therefore, the focus on discovering possible AhR inhibitors may represent a successful strategy.

The models here discussed represent the first multidomain model of the human AhR involving three domains, excluding those automatically generated by AI systems such as AlphaFold that covers the whole protein sequence. We decided not to use it in our docking study due to its unsatisfactory quality. In fact, the lack of suitable templates for each of the protein domains, even for AlphaFold, means that predictions on the whole structure can only have low levels of confidence, although regions related to the PAS A and PAS B domains show a degree of confidence within the very high range. The region linking these two domains was predicted again with confidence between the low and very low ranges (Figure 2A). Therefore, it is possible to assert that AlphaFold does not give a confident resolution of how these different domains combine with each other and the whole structure remains not predicted, in terms of domain assembly. In fact, the AlphaFold model, as expected, shows a lower quality value (see Appendix A) with a ProSA Z-score = −6.14 but a local model quality plot positive for half of the structure and 73% of amino acids in the allowed region of the Ramachandran plot. However, the evaluation of the first three domains (bHLH, PAS A, and PAS B until the PAC motif) shows better parameters for these regions, comparable to the ones related to our multidomain model (see Appendix A). From the overlap of the two models, it is evident that the secondary structures of the PAS A and PAS B domains are quite the same (Figure 2B,C) (RMSD: 1.469 and RMSD: 1.040, for the first and the last, respectively). However, the assembly of the molecule is different, with a different orientation of the PAS A and B domains. The good comparability of the two models, and the very low prediction confidence score related to the region linking the two domains, cannot suggest a better folding of the AlphaFold model than the one manually modeled. Based on these considerations, the latter was recognized as being accurate enough for our docking studies.

Several natural compounds were investigated for their ability to modulate AhR activity, but often the exact mechanism(s) of action remained unknown. In the present study, we focused our attention on compounds of different phytochemical classes aiming to find evidence of their potential effects on AhR signaling that could be translated into reducing the hazardous exposure to PHAs.

According to our data, carotenoids represent the most interesting compounds potentially able to inhibit the AhR-ARNT binding. Lycopene and β-carotene are indicated in the literature to be promising AhR inhibitors that are able to protect from toxicant effects in vitro and in vivo models [14,15,27,28]. The experimental validation (Figure 5) confirmed the inhibitory effect of β-carotene, which at low concentrations (1 μM) significantly reduced the expression of the target gene CYP1A1. However, there are only few studies relating to the action of carotenoids on the AhR; furthermore, they do not refer to the exact mechanism of action of these compounds. The lack of information may be probably due to the difficulty of working with this poorly soluble class of compounds. Our results, however, open new interesting perspectives, providing the hypothesis of a novel inhibition mechanism related to the heterodimerization hindrance. In fact, due to the molecular size of β-carotene, the interactions of this compound with the AhR involve residues of both the PAS A (F82-F83-G96-G97-Q98-V126-V128-F136-Y137-C219-Q240-G241-F266-A267) and PAS B domains (D329-Y332-I349-V350) stabilizing a possible closer conformation of these two regions, which in this arrangement are not able to bind ARNT. Moreover, the location of the β-carotene in the PAS A domain (pocket D) directly covers the ARNT:AhR heterodimerization interface. This could explain the reduced CYP1A1 activation in cells treated with BaP and β-carotene in combination, where β-carotene is probably preventing the ARNT binding, counteracting the BaP-induced AhR translocation into the nucleus. It is worthwhile to note that the results shown in Figure 5A on the protective effect of β-carotene in response to BaP treatment can appear underestimated. In fact, a corollary of the docking analyses reported in Table 2 and Appendix A is that β-carotene, or at least a sub-fraction of it, must be present inside the HepG2 cells as an unmetabolized compound. However, it is well known that the intracellular metabolism of β-carotene is mediated by the scavenger receptor class B type 1 (SR-B1) followed by the action of carotenoid cleavage dioxygenases (CCDs or BCO1-2) that catalyze a region-specific oxidative cleavage in the central polyene chain of carotenoids converting them into more polar metabolites (reviewed in [29]). Therefore, we can hypothesize the presence of a pool of uncleaved molecules able to bind and inhibit AhRs whose concentration will be bona fide lower than the Ki expected, based on the docking simulations results, and certainly significantly lower than the one initially applied to the cells. This hypothesis, if correct, may also justify why the inhibitory effect of β-carotene shown in Figure 5A, although statistically significant, is limited to about a 25% decrease with respect to CYP1A1 expression following BaP treatment. In future studies, it will be essential to study the bioavailability of β-carotene in HepG2 and other cell lines using the same treatment here reported to assess its intracellular concentrations dose- and time-dependent.

Concerning the other classes of compounds, the “borderline” results of the docking reflect the controversial data present in the literature. Among the class of flavonoids, compounds such as quercetin and naringenin were indicated both as agonist or antagonist ligands of AhRs [30,31,32,33]; of course, this dual and controversial activity must be due to dosages applied and cellular models employed. Docking results related to quercetin support this, as this flavonol seems to be a low-affinity ligand for the AhR, explaining in part also the rise in the CYP expression level in cells treated with quercetin alone and in combination with TCDD [34]. Our results highlight that this compound has no higher selectivity for inhibition pockets, showing a possible weak interaction at the activation one.

Concerning curcumin, data suggest its ability to modulate AhR signaling. In the literature, for example, it was described that the molecule attenuates AhR/ARNT-mediated CYP induction by dioxin through the degradations of the AhR and ARNT [35]; curcumin binds the AhR as a ligand but acting on PKC inhibits its transformation [36]. A recent publication showed that curcumin facilitates the AhR activation to regulate inflammatory astrogliosis, enhancing the inflammation-induced IDO/KYN axis and allosterically regulating endogenous ligand binding to the AhR [37]. However, our docking results do not reveal a direct effect on AhR binding. Despite the detection of good binding energy for a possible interaction at the region between the PAS A and PAS B domains, to be relevant for a possible closer conformation of the receptor and for a possible modification of the underlying pocket C, no selectivity for this pocket has been detected. These data are in line with those shown in Figure 5B that do not evidence significant differences in cells treated with BaP alone with respect to the combined treatment of BaP plus curcumin.

The protective effect of the ellagic acid against TCDD stress was already observed in the literature, above all, for what concerns the CYP1A1 activity in mice models pre-treated with this compound [38]. In agreement with this observation, our data support the conclusion that the combined treatment of ellagic acid plus BaP reduces CYP1A1 expression (Figure 5C). The docking results retrieved for this compound are not easy to interpret. Ellagic acid seems to have a possible inhibitory role binding the area between pockets A and B with a strong selectivity (30 poses in the cluster) and a discrete binding energy (MBE = −6.77 kcal/mol; LBE = −7.28 kcal/mol). This result is supported by a number of additional clusters whose poses, from our analysis, are still compatible with a potential inhibitory role. However, in the 11th cluster in the blind docking rank, with an MBE equal to −6.23 kcal/mol and 12 poses in the cluster, pocket C is present, suggesting a possible activation role at a high concentration. Nevertheless, the docking simulations show that the selectivity and the binding energy of ellagic acid were comparable to those detected for TCDD, while the latter was higher of about −2.00 kcal/mol, with respect to the one measured for BaP.

The hypothetical mechanism of inhibition may reside in the block of the heterodimerization, explaining a lower response of the AhR, which on one side preserves its ability to bind BaP that, at the xenobiotic binding site (pocket C), has a better binding affinity than ellagic acid, but on the other, reduces its translocation activity. Moreover, even the detection of a possible binding in pocket C that is, in appearance, in contrast with AhR inhibition could be considered a possible inhibitory effect. Several studies highlighted the PAS B domain, and that pocket C in general can be divided into at least three pockets [39], which could be responsible for a different response of the AhR. Actually, agonist and antagonist compounds bind the same pocket C in different positions making stronger interactions with distinct residues [34]. Ellagic acid in our docking simulation binds pocket C in an upper region that is different from the one detected for BaP or TCDD, disclosing the hypothesis that this binding could have a possibly more divergent effect than the canonical activation.

Although partially speculative, an alternative interpretation can explain the controversial data obtained for ellagic acid. As reported above in the Results section, the activation of the receptor following ellagic acid binding to pocket C may occur when the molecule is present at high concentrations, while the inhibitory response mediated by the ellagic acid binding to pockets A and B more likely requires a lower concentration. This observation can match the comments reported above concerning β-carotene bioavailability and intracellular metabolism. In fact, old data indicate that in mouse primary keratinocytes treated with ellagic acid and BaP, a large amount of ellagic acid was sequestered by organic (BaP-7,8-diol, 3-OHBaP, BaP-9,10-diol) and water-soluble BaP conjugates (3-OH-BaP, several BaP-quinones, and BP-7,8-diol conjugated with glucuronide and sulfate, respectively) present both intracellularly and in the extracellular culture medium [40]. Therefore, it is plausible that free and unmetabolized ellagic acid inside the cells is present at a concentration not high enough to bypass the threshold necessary to bind pocket C and activate the receptor, while it is sufficient to guarantee the inhibitory effect confirmed by the in vitro assay reported in Figure 5C. In this case, also, new evidence will derive from the determination of the intracellular concentration of ellagic acid.

In view of the achieved results, it is noteworthy to underline that we used the biomarker CYP1A1 as an indicator of the AhR activation; however, the biological and functional validation of the information resulting from this study will be deepened in the future, taking into account also the possible impact/findings in different fields of investigation. The broad spectrum of functions related to the AhR attracts the interest of a diverse audience of scientists from environmental to pathophysiological areas. Counteracting the canonical pathway of AhR activation, which is essential in mediating the toxic effect of contaminants but also in cancerogenesis processes, is an extremely interesting topic. Recently, the search for selective AhR modulators (SAhRMs) became a growing field of investigation following the concept introduced by Safe et al., according to which a receptor–ligand exhibits a tissue-specific AhR agonist or antagonist effect [41,42,43,44]. However, the development of SAhRMs presents an important challenge because the selectivity of the response, agonist vs. antagonist, cannot be easily predicted. Thus, many bioassays, considering also differences in tissue/organ/species-specific responses, are needed to identify the optimal ligand for specific clinical applications. In this context, the developed AhR model here presented may certainly represent an opportunity.

## Figures and Tables

**Figure 1 foods-12-01953-f001:**
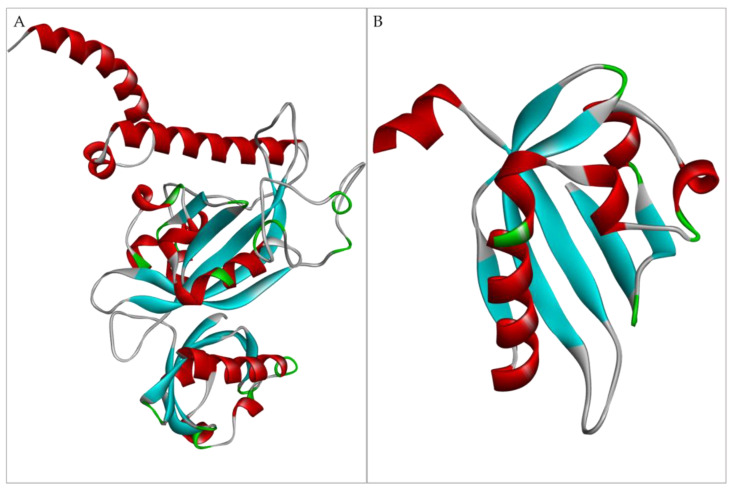
AhR models. Representation in cartoon, colored by secondary structure, of: (**A**) the bHLH, PAS A, and PAS B (until the PAC motif) domains model of AhR; (**B**) the PAS B domain model of AhR built by Modeller 9.19.

**Figure 2 foods-12-01953-f002:**
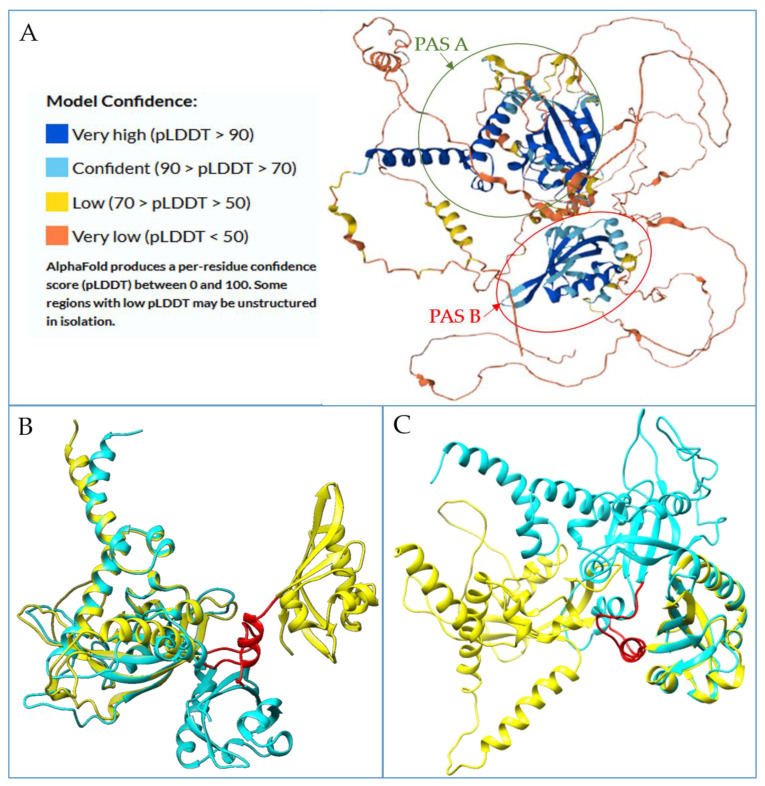
Comparison between the AlphaFold model and the multidomain model built manually: (**A**) AlphaFold model represented in cartoon and colored by levels of confidence, the PAS A and PAS B domains highlighted by circles; (**B**) overlap of the PAS A domains of AlphaFold model in yellow cartoon and the manual one in cyan cartoon; (**C**) overlap of the PAS B domains. In red, the alpha helix linking the two domains in AlphaFold model.

**Figure 3 foods-12-01953-f003:**
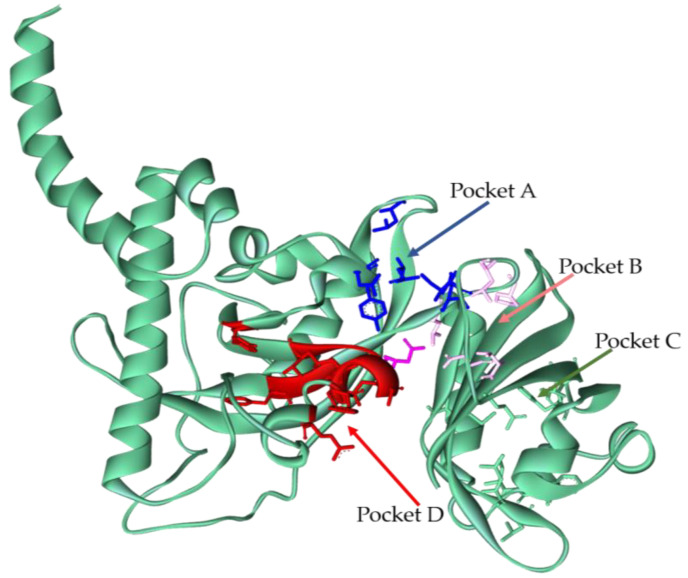
Pockets detected by ligands on AhR structure. Representation in green cartoon of AhR; residues that make up the pocket are represented in stick: blue for pocket A, pink for pocket B, green for pocket C, and red for residues of pocket D, involved in ARNT interaction. In magenta, stick N123, a residue located in PAS A domain, that is involved in all three pockets: A, B, and D.

**Figure 4 foods-12-01953-f004:**
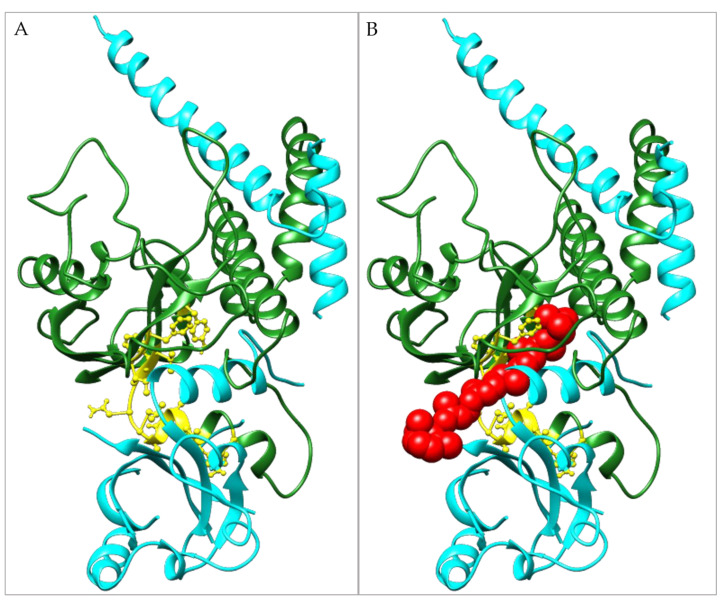
AhR:ARNT heterodimerization. Representation in cartoon of: (**A**) the crystallographic structure of AhR PAS A domain in green and ARNT in cyan (from PDB: 5nj8 chains A and B), in yellow, stick AhR residues relevant for the binding to ARNT; (**B**) in green, carton structure of the modeled PAS A domain of AhR and the crystallographic structure of ARNT (complex obtained by the overlap between 5nj8 chain A and AhR multidomain model), in red spheres, β-carotene bound to AhR which takes up the ARNT binding area, as demonstrated by the visible clash between its atoms and ARNT alpha helix.

**Figure 5 foods-12-01953-f005:**
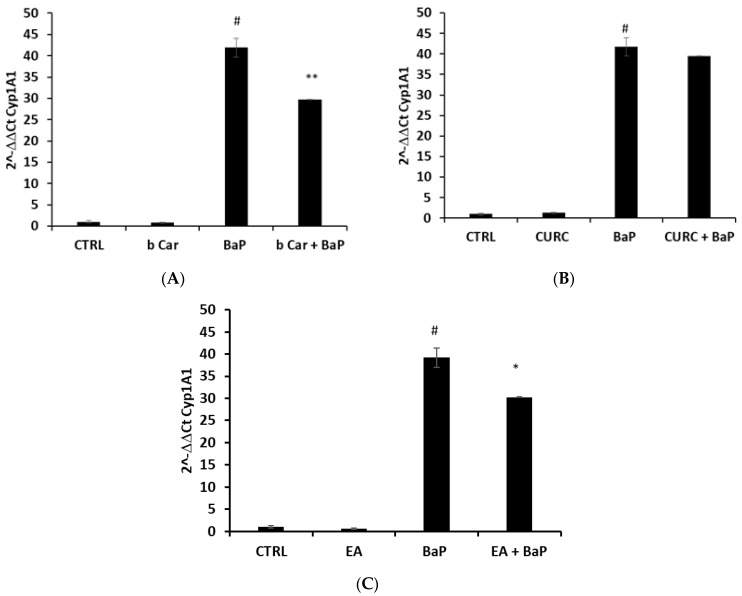
Effect of (**A**) β-carotene, (**B**) curcumin, and (**C**) ellagic acid against BaP-induced over-expression of CYP1A1. HepG2 cells were treated for 1 h with selected compounds (1 μM), followed by addition of BaP (5 µM) for 15 h. Real-Time PCR was performed to assess CYP1A1 mRNA level. Bar graphs indicate the value expressed as 2^−ΔΔCt^ ± SD. Symbols indicate significance with: # *p* < 0.001 respect to CTRL; * *p* < 0.05 and ** *p* < 0.005 compared to BaP single treatment.

**Table 1 foods-12-01953-t001:** Table of docking results for control molecules.

Docking Type	Ligand	Binding Energy (kcal/mol)	Estimated Inhibition Constant (Ki)	N° of Poses inCluster	Pocket
FOCUSED	**TCDD**	−7.07	6.07 µM	49	C
		−5.75	60.40 µM	29	Left side of B
BLIND		−7.75	2.05 µM	27	D
		−7.75	2.07 µM	49	A
		−7.49	3.07 µM	21	PAS1
PARTIALLY FOCUSED		−7.70	2.24 µM	67	A
		−7.50	2.37 µM	19	B
		−6.57	14.87 µM	8	C
BLIND	**BaP**	−9.22	147.01 nM	12	C
		−8.92	282.51 nM	57	A
		−8.44	598.05 nM	3	B
		−8.00	1.34 µM	8	D
FOCUSED		−9.33	112.85 nM	40	C

**Table 2 foods-12-01953-t002:** Ligand Ranking. Ligands are ranked in order of major probability of inhibiting the AhR. The table reports: the ranking position, the ligand, the position of the selected results among all the docking results (Cluster rank), the binding energy measured in kcal/mol (MBE: the mean binding energy of all the cluster; LBE: the lowest binding energy of the cluster), the number of poses present in each cluster (a higher number corresponds to a higher selectivity), and the binding pocket detected on AhR multidomain model.

Rank	Ligand	Cluster Rank	Binding Energy (kcal/mol)	N° in Cluster	Pocket
1	β-carotene	1	MBE = −10.65 (LBE = −12.57)	32	D
2	Lycopene	1	MBE = −8.24 (LBE = −9.96)	4	between A and B
		2	MBE = −0.84 (LBE = −9.57)	12	D
3	Xanthohumol	2-3-4-6	lowest MBE = −9.05	2 (+8)	between A and B
		7	MBE = −6.68 (LBE = −7.14)	7 (+5)	D
4	Hesperetin	1	MBE = −7.75 (LBE = −8.10)	8	A
		2	MBE = −7.49 (LBE = −8.02)	2	C
		3	MBE = −7.44 (LBE = −8.00)	7	D
5	Naringenin	1-2-8	lowest MBE = −7.94	2 (+16 +7)	A
		4-5	lowest MBE = −7.62	9 (+3)	B
		9	MBE = −7.04 (LBE = −7.30)	5	C
		18-19-21	lowest MBE = −6.83	11 (+1 +4)	D
6	Curcumin	4-5-6-7	lowest MBE = −7.87	2 (+2 +1 +1)	between A and B
		13-14	lowest MBE = −7.10	1 (+2)	A
		17-18	lowest MBE = −6.50	3 (+3)	D
7	Quercetin	1-2-5	lowest MBE = −7.46	2 (+9 +3)	A
		3	MBE = −7.21 (LBE = −7.41)	6	B
		6	MBE = 6.94 (LBE = −7.19)	4	between A and B
		9	MBE = −6.41 (LBE = −6.68)	14	D
		10	lowest MBE = −6.65	2	C
8	Resveratrol	1	MBE = −7.33 (LBE = −7.82)	11	B
		5-6-8-15	lowest MBE = −6.71	11 (+3 +11 +4)	A
		7-14	lowest MBE = −6.54	3 (+3)	D
9	Cyanidin-3-glucoside	5-7-11	lowest MBE = −5.70	1 (+2 +1)	D
10	Epigallocatechin-3-gallate	3-9	lowest MBE = −7.93	1 (+5)	D
		6	MBE = −6.65 (LBE = −7.58)	2	B
11	Genistein	4	MBE = −6.82 (LBE = −7.13)	17	A
		6	MBE = −6.84 (LBE = −7.07)	9	between A and B
		12	MBE = −6.57 (LBE = −0.83)	9	C
		21	MBE = −6.08 (LBE = −6.11)	3	D
12	Hydroxytyrosol	2-8-17	lowest MBE = −4.88	2 (+5 +3)	B
		4-13-14-19-22-23	lowest MBE = −4.90	2 (+4 +3 +4 +5 +3)	A
		7-9	lowest MBE = −4.79	2 (+3)	D
13	Ellagic acid	1-2	lowest MBE = −6.77	30 (+5)	between A and B
		4	MBE = −6.59 (LBE = −7.28)	7	A
		8	MBE = −6.59 (LBE = −7.02)	4	B
		11	MBE = −6.23 (LBE = −6.24)	12	C
14	Sulforaphane	1-2-3	lowest MBE = −5.68	1 (+3 +1)	B
		5-10-18-23-26	lowest MBE = −4.91	9 (+6 +4 +3+4)	C
		8-11-24	MBE = −4.42 (LBE = −4.82)	4 (+7 +2)	A

## Data Availability

Data presented in this study are available on request from the corresponding author.

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
