# Peer review of "Molecular Docking of Natural Compounds for Potential Inhibition of AhR"

_foods, 2023, doi:10.3390/foods12101953_

Round 1

Reviewer 1 Report

The docking studies presented in this manuscript go well beyond relatively straightforward mix-and-match docking studies that are regularly reported to support enzyme inhibition by natural products. The authors carefully explain and justify their choices of target enzymes and ligands. The data result in several credible hypotheses that explain previous observations that have been made on naturally occurring AhR inhibitors.

It would be helpful if in Figure 2A the positions of the PAS A and the PAS B domains are indicated. This would allow the reader to more easily compare the data presented in figures 2B and 2C

The experimental data obtained with in vitro experiment support the hypotheses derived from the docking studies. The manuscript brings us closer to understanding the different ways in which natural compounds that are part of our diet contribute to regulation of AhR expression.

I only have some minor suggestions for corrections:

Line nr

22       Two of the compounds retrieved from the docking simulations, i.e.  β-carotene and ellagic acid, confirmed...

25       benzo[a]pyrene (see also lines 27, 36, 90)

59       In contrast, its ability to interact with members of the bHLH-PAS class II is well known;...

61       ... in particular, with the aryl hydrocarbon receptor nuclear translocator...

65       ...the entire complex can then translocate...

75       ...flavone, flavonol, flavanone, and isoflavone subclasses...

77       ...range of concentrations...

176      CYP1A1 (CYP capitalized) (see also: 347, 351, 355, 356, 401, 412, 428, 458, 500, 593) In addition, it is customary to write names of genes in italics and the names of the corresponding proteins in normal font.

259      kcal (with a lower case’k’) (see also tables and 269, 270, 271, 293, 299, 304, 314, 325, 326, 328, 462, 465, 468).

370      ...due to its unsatisfactory quality.

371      ... makes that predictions on the whole structure can only have a low level of confidence

436     Docking results related to quercetin support this, as this flavonol seems to be...

Author Response

The authors would like to thank R1 for his/her positive evaluation of our manuscript. Attached our reply

Reviewer 2 Report

The manuscript describes an essential study on estimating a potential AhR inhibition in presence of well known natural compounds. Three compounds were tested to inhibit the BaP-induced activation of AhR. The authors well combined the in silico approach with confirmation in an in vitro cellular model. Thus the report deserves to be evaluated in its present form.

A minor comment: - why only one time duration (= 15h) of BaP pretreatment in HepG2 cells was selected instead of testing of at least 2-3 different time intervals? The use of additional time intervals of treatment could bring some new information on inhibition of AhR-mediated pathway in presence of b-carotene, or ellagic acid. It may be useful to discuss this point of view in the discussion - but not necessary.

Author Response

The authors would like to thank R2 for his/her positive evaluation of our manuscript. Attached our reply
